# Computational Analysis of Short Linear Motifs in the Spike Protein of SARS-CoV-2 Variants Provides Possible Clues into the Immune Hijack and Evasion Mechanisms of Omicron Variant

**DOI:** 10.3390/ijms23158822

**Published:** 2022-08-08

**Authors:** Anjana Soorajkumar, Ebrahim Alakraf, Mohammed Uddin, Stefan S. Du Plessis, Alawi Alsheikh-Ali, Richard K. Kandasamy

**Affiliations:** 1College of Medicine, Mohammed Bin Rashid University of Medicine and Health Sciences, Dubai P.O. Box 505055, United Arab Emirates; 2Dubai Health Authority, Dubai P.O. Box 4545, United Arab Emirates; 3Centre of Molecular Inflammation Research (CEMIR), Department of Clinical and Molecular Medicine (IKOM), Norwegian University of Science and Technology, 7491 Trondheim, Norway

**Keywords:** coronaviruses, SARS-CoV-2, variant, Omicron, SLiMs, spike protein, motifs, COVID-19

## Abstract

Short linear motifs (SLiMs) are short linear sequences that can mediate protein–protein interaction. Mimicking eukaryotic SLiMs to compete with extra- or intracellular binding partners, or to sequester host proteins is the crucial strategy of viruses to pervert the host system. Evolved proteins in viruses facilitate minimal protein–protein interactions that significantly affect intracellular signaling networks. Unfortunately, very little information about SARS-CoV-2 SLiMs is known, especially across SARS-CoV-2 variants. Through the ELM database-based sequence analysis of spike proteins from all the major SARS-CoV-2 variants, we identified four overriding SLiMs in the SARS-CoV-2 Omicron variant, namely, LIG_TRFH_1, LIG_REV1ctd_RIR_1, LIG_CaM_NSCaTE_8, and MOD_LATS_1. These SLiMs are highly likely to interfere with various immune functions, interact with host intracellular proteins, regulate cellular pathways, and lubricate viral infection and transmission. These cellular interactions possibly serve as potential therapeutic targets for these variants, and this approach can be further exploited to combat emerging SARS-CoV-2 variants.

## 1. Introduction

SLiMs are short conserved sequences approximately 3–10 amino acids long with distinct functions that typically lie in the disordered regions of proteins [1]. The characteristic features of the SLiMs are that a single sequence can recognize multiple proteins via a small contacting surface, and any substitution or modification of a single residue significantly modifies its binding affinity to the proteins [2]. Moreover, the short length of the motifs and these factors brings forth an adaptable molecular basis for the rapidly evolved proteins of RNA viruses to develop high versatility.

Viruses mimic cellular linear motifs and hijack host cell factors, favoring viral infection. Severe acute respiratory syndrome coronavirus 2 (SARS-CoV-2) is a single-stranded, positive-sense RNA virus, and its evolution has been of great concern since the beginning of the pandemic [3,4]. The SARS-CoV-2 spike protein (S-protein) exists as a trimeric protein. It consists of the S1 subunit with the N-terminal domain (NTD), receptor-binding domain (RBD), subdomains 1 and 2 (SD1 and SD2), and the S2 subunit, with heptad repeat 1 (HR1), the central helix (CH), a connecting domain (CD), heptad repeat 2 (HR2), a transmembrane (TM) domain, and the C-terminal part region [5]. The S-protein equips the virus to enter the host cells and is the hotspot of viral evolution and the fundamental attribute in developing many vaccines [6].

Multitudinous factors are involved in the pathogenesis of SARS-CoV-2 infection and are highly complex. In addition, the continuous evolution of SARS-CoV-2 and the mutations specifically in the spike protein give rise to variants having unique characteristics, such as the altered transmissibility and severity of the disease. The SARS-CoV-2 Omicron variant has become pronounced and highly transmissible, and is spreading faster than any previous variant [7,8]. Upon infection, viruses trigger a cascade of antiviral responses in the host cells, and they must develop mechanisms to evade and subvert those antiviral responses. SLiMs in viral proteins manipulate the host cell’s essential processes, such as cell signaling, cell cycle, DNA damage repair, and immune system [9], to escape from the responses. Therefore, unraveling the system by which viruses modify these processes can promote the expansion of new rational therapies.

We performed an amino acid sequence analysis across the SARS-CoV-2 variants’ spike protein for potential functional linear motifs using the eukaryotic linear motif (ELM) resource. The principal question was to discover the interference of viral SLiMs with human host networks. In this study, we identified several SLiMs on the spike protein unique to SARS-CoV-2, and the landscape of SLiMs across all SARS-CoV-2 variants. This could potentially uncover the possible functions of SLiMs identified in the Omicron variant taking advantage of known host motif–protein interactions, assuming that the recognized motifs have common interactors in the host, and exploiting the pathways for viral entry, replication, and dissemination. These findings can be crucial to developing therapeutic intervention strategies that are essential to contain emerging SARS-CoV-2 variants.

## 2. Results

### 2.1. Sequence Analysis Shows Unique and Common SLiMs to SARS-CoV-2 Variants

We performed sequence analysis to identify SLiMs across the various coronaviruses’ spike protein sequences (Figure 1A), and found six SLiMs unique to SARS-CoV-2 as compared to other coronaviruses, namely, LIG_Integrin_RGD_1, LIG_CaM_IQ_9, MOD_CMANNOS, LIG_GBD_Chelix_1, LIG_AP2alpha_2, and LIG_WW_1 (Appendix A). For instance, several studies showed the presence of LIG_Integrin_RGD_1 in SARS-CoV-2 due to a single amino acid mutation, and how this impacts diverse cellular processes including cell entry [1,2]. These SLiMs potentially confer SARS-CoV-2 with numerous mechanisms to manipulate the host immune response. Since multiple studies have investigated the occurrence and impact of novel SLiMs associated with the SARS-CoV-2 spike protein, this study focused on the SLiMs across the various variants of interest and concern described by the Centers for Disease Control and Prevention (CDC; Appendix A).

With the SARS-CoV-2 Omicron variant containing 34 different mutations compared to the first Wuhan variant (Figure 1B and Appendix A), our sequence analysis for the occurrence of SLiMs identified 92 motifs in the Omicron variant, and 88 motifs in both the Delta and Wuhan variants (Figure 1C,D). There were five SLiMs that were unique to the Omicron variant as compared to the Wuhan variant, namely, LIG_TRFH_1, LIG_REV1ctd_RIR_1, LIG_CaM_NSCaTE_8, MOD_LATS_1 and MOD NEK_2 (Figure 1C). Here, we discuss only four motifs, and the summary of the ELM accession data of those four identified SLiMs is shown in Table 1. The multiple sequence alignment of the spike protein from various SARS-CoV-2 variants resulted in four subgroups where the Omicron, Delta, Iota, and Kappa variants are closely related, clustering into one group (Figure 1E). Overall, this analysis shows both unique and common SLiMs across coronaviruses and across SARS-CoV-2 variants, which would play potential roles in modulating and hijacking the host immune response.

### 2.2. LIG_TRFH_1 Motif Interaction with Shelterin Components -TRF2 and TIN2

We recognized the existence of an LIG_TRFH_1 motif (Figure 2A,B) in the SARS-CoV-2 Omicron and Iota variants. In humans, the telomere repeat factor (TRF) homology (TRFH) domain interacts with the conserved TRFH-binding motif (TBM) in the telomere repeat factor 1 (TRF1), telomere repeat factor 2 (TRF2), and TRF1-interacting nuclear protein 2 (TIN2) components of the shelterin complex [10]. Shelterin, a complex of six proteins, TRF1, TRF2, POT1, TPP1, TIN2, and Rap1, protects telomeres from various types of DNA damage and prevents end-to-end chromosomic damage fusions [11]. Figure 2C represents the structural organization and interaction of human TRF1, TRF2, and TIN2. These proteins and other binding partners of shelterin protect telomeric DNA sequences (Figure 2D). The depletion or mutation of any of the components of the shelterin complex triggers telomeric dysfunction and activates DNA damage response (DDR) pathways [12].

Viral infection can also induce DNA damage and evoke the host DDR, and several studies reported that the progression of SARS-CoV-2 severity is associated with decreased telomere length [13,14]. A study on African green monkey kidney cells (Vero E6) showed that SARS-CoV-2 infection could trigger DNA damage response (DDR) and impact telomeric stability [10,15]. We assumed, analogously to cellular TRFH, the identified viral LIG_TRFH_1 could interact with TRF2 and TIN2 proteins. A possible interaction between LIG_TRFH_1, and TRF2 and TIN2 is presented in Figure 2E. TIN2 is the shelterin complex’s central hub that interacts with TRF1, TRF2, and TPP1, and promotes the assembly of an intact shelterin complex that protects the telomeric repeats. TRF2 participates in t-loop formation and protects telomeres from DDR pathways [16].

We presumed that, in Omicron-infected cells, this interaction might favor the assembly of the other shelterin components at the viral terminal repeats, and thus establish a protective complex that can effectively suppress the cellular DDR, as shown in Figure 2F. Furthermore, we assumed that, besides evading the host immune attack, the LIG_TRFH_1 SLiM in Omicron might also indirectly favor maintaining the host’s telomeric stability to protect the cells from senescence. Therefore, LIG_TRFH_1 could accelerate viral propagation and cellular replication without any hurdle; thus, the participation of the LIG_TRFH_1 motif in these two events might diminish the severity of Omicron infection.

### 2.3. LIG_PCNA_TLS_4 and LIG_REV1ctd_RIR_1 Motifs Are Associated with the Mutagenic TLS

We located a potential LIG_REV1ctd_RIR_1 motif (Figure 3A,B) exclusively in the spike protein of the SARS-CoV-2 Omicron variant. This SLiM contains a functional RIR motif that is present in the Y-family polymerases that play a central role in translesion DNA synthesis (TLS), a mutagenic branch of cellular DNA damage tolerance [17,18]. The RIR motif has been identified in the Y-family DNA polymerases Polη, Polι, and Polκ, and it interacts with the C-terminal domain of Rev1. Rev1 is a crucial member of eukaryotic Y-polymerases, and its structure is shown in Figure 3C. Rev1 recruits inserter and extender polymerases to the lesion site through specific protein–protein binding via its C-terminal domain. Furthermore, the systematic action of these TLS polymerases is controlled through their interactions with the two scaffold proteins, the sliding clamp Proliferating cell nuclear antigen (PCNA) and the TLS polymerase Rev1. Rev1 also has an N-terminal BRCT domain by which it interacts with the PCNA.

In addition to the LIG_REV1ctd_RIR_1 motif, we also tracked down motif LIG_PCNA_TLS_4 expressed in all the SARS-CoV-2 variants. The monoubiquitination of PCNA promotes TLS machinery and serves as a docking site for key TLS players. Figure 3D depicts the recruitment and interaction of mammalian TLS-associated proteins via PCNA and Rev1. The switch between the replication polymerases and TLS polymerases relies on the post-translational modification status of PCNA. Recently, increased ubiquitination in specific regions of PCNA was shown in SARS-CoV-2-infected cells compared to the control group, indicating a regulatory mechanism of PCNA during viral infection [19,20]. This can be interpreted as viral-infection-induced damage, potentially mediating the monoubiquitination of PCNA that promotes TLS with increased mutagenesis. We hypothesized that the LIG_PCNA_TLS_4 triggers the initiation of the TLS events in Omicron, and the interaction of SLiM LIG_REV1ctd_RIR_1 with the cellular Rev1 recruits its associated proteins and initiates the mutagenic TLS mechanism (Figure 3E,F).

### 2.4. Ca^2+^/CaM Mediated Binding of LIG_CaM_NSCaTE_8 and LIG_CaM_IQ_9 SLiMs

We discovered a pre-eminent motif, LIG_CaM_NSCaTE_8 (Figure 4A), in the Alpha, Eta, and Omicron variants of SARS-CoV-2. Previously, Ben-Johny et al. proposed a blueprint to show the involvement of N-terminal spatial calcium transforming element (NSCaTE) in calcium (Ca^2+^)/calmodulin (CaM)-mediated regulation of voltage-gated calcium (Cav) channels based on an NMR structure [21,22]. In the resting state, CaM interacts with an IQ motif via its C-lobe, and the NSCaTE motif preferentially interacts with an N-lobe of CaM and induces a conformational change (Figure 4C) [23]. The presence of NSCaTE increases the gross channel affinity for Ca^2+^/CaM over Ca^2+^-free CaM (apoCaM), providing Ca^2+^ selectivity, and the elimination or donation of NSCaTE regulates the spatial Ca^2+^ selectivity between global and local profiles.

Interestingly, we also identified a LIG_CaM_IQ_9 (Figure 4B) motif in all the variants of SARS-CoV-2. NSCaTE can also interact with Ca^2+^/CaM prebound to an IQ domain peptide, suggesting a possible amino- and carboxyl-terminus bridging of the channel [24]. When the two viral motifs of LIG_CaM_NSCaTE_8 and LIG_CaM_IQ_9 (Figure 4A,B) come closer in the cytoplasm during Omicron infection, a Ca^2+^/CaM-mediated bridging could occur. A mechanism parallels the Ca_V_1 channels where the NSCaTE element interacts with Ca^2+^/CaM prebound to an IQ domain peptide (Figure 4D). This process depletes the available free cytoplasmic Ca^2+^/CaM level and might slow down the Ca^2+^/CaM-associated mechanisms in virus-infected cells. For instance, Ca^2+^/CaM mediated ACE2 ectodomain shedding, a rate-limiting step for SARS-CoV-2 entry. On this basis, we hypothesized a model explaining the possible involvement of these SLiMs in the shedding processes of ACE2 catalytic ectodomain, which are depicted in Figure 4D.

### 2.5. MOD_LATS_1 Intervention in YAP/TAZ Localization and Hippo Signaling Regulation

The MOD_LATS_1 motif (Figure 5A) was observed in the Alpha, Omicron, Theta, and Kappa variants. In cells, YAP/TAZ are the substrates of mammalian large tumor suppressor homolog (LATS) and are the critical effectors of the Hippo pathway. Cellular LASTS1/2 is activated by phosphorylation, and directly interacts with and phosphorylates YAP/TAZ, which mediates their localization and function in the hippo signaling pathway [25]. TANK binding kinase 1 (TBK1) is crucial for cytosolic nucleic acid sensing and antiviral defense. The Hippo pathway regulates antiviral defense by modulating the TBK1-mediated control of interferon production. Studies show that Hippo activation abrogates the inhibitory effect of YAP/TAZ on TBK1 and enhances antiviral responses [26]. In mammalian cells, LATS1/2-kinase-mediated depletion or deletion of YAP/TAZ relieves TBK1 suppression and boosts antiviral responses. In contrast, the expression of the transcriptionally inactive YAP dampens cytosolic RNA/DNA sensing and weakens cell antiviral defense [27].

Several studies illustrate that YAP/TAZ is a negative regulator of innate immunity against DNA and RNA viruses [28]. We assume that, during Omicron infection, the viral MOD_LATS_1 becomes phosphorylated by the cellular LASTS1/2 and thus controls the Hippo pathway (Figure 5B). YAP/TAZ abolishes the virus-induced activation of TBK1-IKKε and restores viral replication. However, the LASTS1/2-mediated knockout of YAP/TAZ results in enhanced innate immunity and a reduced viral load [29]. The MOD_LATS_1 motif could hijack cellular LATS phosphorylation and thus control the Hippo pathway, suggesting defective YAP/TAZ-related antiviral responses. MOD_LATS_1 becomes phosphorylated by the cellular LASTS1/2, leaving the active cytoplasmic YAP/TAZ in the infected cells. This activity could weaken the antiviral response, allowing viruses to escape immune surveillance.

## 3. Discussion

### 3.1. LIG_TRFH_1 Motif Maintains Viral and Cellular Terminal Repeats from DDR Response

Various viruses utilize different mechanisms for genome end-protection, but none is fully understood. For instance, human herpesvirus 6 (HHV6) and Marek’s disease virus (MDV) contain telomeric repeat sites in viral genomes that can direct the integration of viral genomes towards the host telomere facilitated by TRF1 and TRF2 [30]. In humans, TRFH domains recognize a conserved motif, TBM, present in the shelterin proteins, thereby enabling the integration of communication networks between telomere-associated proteins, DNA-repair, and DNA-damage signaling proteins [10]. The TBM domain is conserved in the TRF2 and TIN2 components of the shelterin.

The genome-wide analysis demonstrates that SARS-CoV-2 has ultra-conserved 59- and 39-terminal regions, which are shared among betacoronavirus lineage B genomes [31]. Furthermore, it has secondary conserved RNA structures that are crucial for the replication and transcription of the virus [32], and can also recruit and interact with a range of host and viral protein factors. This region is susceptible to or detected by the DNA damage-sensing and -repair pathways. We presume that, during Omicron infection, LIG_TRFH_1 can recruit the cellular shelterin proteins to the viral terminal repeats, forming a protective complex that can actively suppress the cellular DDR (Figure 2F). The existence of the LIG_TRFH_1 motif benefits the Omicron variant by potentially hijacking the cellular telomeric factors and mimicking the mechanism by which shelterin suppresses DDR at telomeres. However, further work must be conducted to justify these hypotheses to show the suppression of the cellular DDR by SLiM LIG_TRFH_1. In addition, this motif may also support maintaining host telomeric stability. We hypothesize that the identified viral LIG_TRFH_1 could hijack the cellular TRFH and bind to the TBM motif present in these proteins (Figure 2E).

### 3.2. Generation of Highly Mutagenic Omicron Variant through a Unique LIG_REV1ctd_RIR_1 Motif

The large number of mutations observed in the spike protein of SARS-CoV-2 raised severe concerns about the contagious nature of this new virus with little prior immunity. TLS is a mutagenic branch of cellular DNA damage tolerance that enables bypass replication over DNA lesions accomplished by specialized low-fidelity DNA polymerases [17,18]. Y-family polymerases have a specific preference for incorporating incorrect nucleotides, and the resulting mutations are sequence-specific. LIG_REV1ctd_RIR_1 is exclusively present in the Omicron variant and has a functional RIR motif specific to the Y-family polymerases of TLS. The mammalian RIR motif of polymerases η, ι, and κ can bind the C-terminal domain of Rev1 (Rev1-CTD), which, in turn, interacts with Rev3 and Rev7 subunits of pol ζ (Figure 3D) [33]. In addition, Rev1 has deoxycytidyl (dCMP) transferase activity, and plays a central role in TLS during replication and post-replication DNA repair [34], but lacks polymerase activity.

The interaction between Rev1 and other Y-family DNA polymerases is frail, facilitating rapid and dynamic associations and dissociations among the many factors involved in TLS events. Compared to conventional replicative polymerases, Y-family polymerases have distinct structural and biochemical features that bypass DNA damage. Figure 3E represents the possible participation of the LIG_PCNA_TLS_4 and LIG_REV1ctd_RIR_1 motifs in TLS. The presence of LIG_PCNA_TLS_4 may trigger the initiation of the TLS events in Omicron, and the unique LIG_REV1ctd_RIR_1 interacts with the cellular Rev1, recruits its associated proteins, and initiates the mutagenic TLS mechanism. Figure 3F illustrates the involvement of components of TLS in viral mutagenesis.

The report shows that Omicron has at least 50 mutations, with 35 of those mutations turning up in the spike protein (Figure 6) compared to the reference strain [35]. We suspect that the two SLiMs, LIG_PCNA_TLS_4, and LIG_REV1ctd_RIR_1, promote mutagenesis in Omicron-infected cells mainly by participating in crucial protein–protein interactions that regulate the access of translesion DNA polymerases to the primer terminus. Future investigations must be performed to prove these observations, and explore the involvement of SLiMs LIG_PCNA_TLS_4 and LIG_REV1ctd_RIR_1 in the regulation of the TLS mechanism. Studies indicate that the mutations that emerged in the new SARS-CoV-2 Omicron variant are an essential driver of its increased transmissibility and are supported by our observations.

### 3.3. Dynamic Switching of Cytoplasmic Ca^2+^/CaM Interaction and Inhibition of ACE2 Ectodomain Shedding

The viral proteins expressed on the surface of the infected cells are capable of perturbing intracellular Ca^2+^ homeostasis in several ways [36]. In Omicron-infected cells, an elevated Ca^2+^ in the cytoplasm expedites the Ca^2+^/CaM-mediated bridging of the two viral motifs, LIG_CaM_NSCaTE_8 and LIG_CaM_IQ_9, a mechanism analogous to the Cav1 calcium channels where NSCaTE interacts with Ca^2+^/CaM prebound to an IQ domain peptide [24]. We propose a model demonstrating the involvement of two motifs in ACE2 receptor catalytic ectodomain shedding (Figure 4D).

SARS-CoV-2 enters the cells by binding to ACE2 receptors that comprise an extracellular heavily N-glycosylated N-terminal domain containing the carboxypeptidase site and a short intracellular C-terminal cytoplasmic tail. This protein exists in two forms: a membrane-bound cellular form through which the virus enters the cell, and a soluble circulating form [37,38] where SARS-CoV-2 can bind but cannot duplicate because of the unfavorable environment. Hence, the circulating ACE2 is presumed to protect from SARS-CoV-2 infection. Circulating ACE2 is cleaved from the full-length ACE2 on the cell membrane by disintegrin and metalloproteinase domain-containing protein 17 (ADAM17), and then liberated into the extracellular environment. The calcium signaling pathway is involved in the catalytic ectodomain shedding process of the ACE2 regulated by CaM. Computational analysis of the cytoplasmic domain of ACE2 revealed a conserved consensus calmodulin-binding motif, and the calcium-dependent CaM–peptide complex could bind to the cytoplasmic domain of ACE2 and enhance the ADAM17-mediated ectodomain shedding [39,40].

The interaction of the LIG_CaM_NSCaTE_8 and LIG_CaM_IQ_9 motifs could potentially lead to a depletion in the available CaM for ACE2, and the calmodulin-binding motif in the cytosolic tail of ACE2 remained unoccupied. The absence of CaM downregulates the ectodomain shedding, and provides full-length ACE2 for viral binding and entry. This process results in increased active ACE2 expression in the cell surface, and this could be one of the possible reasons for the high transmissibility of the SARS-CoV-2 Omicron variant. This hypothesis should be further tested to determine the role of SLiMs LIG_CaM_NSCaTE_8 and LIG_CaM_IQ_9 in the shedding processes of ACE2. However, our finding is further supported by recent reports that Omicron infection is ACE2-dependent, and that the binding of the Omicron spike to ACE2 is elevated compared to the wild-type virus [41,42].

### 3.4. Omicron Escapes Immune Surveillance by MOD_LATS_1 SLiM

The Hippo components’ expression and activity serve as an indicator to regulate the magnitude of host antiviral responses. YAP/TAZ mainly functions as a transcriptional coactivator that regulates the transcription of target genes by shuttling between the nucleus and the cytoplasm, thereby affecting cellular immune surveillance against pathogen attacks. In response to viral nucleic acids, YAP blocks the activation of IRF3 at the dimerization step, and limits IFN-β expression and innate antiviral responses to viruses [43].

Omicron MOD_LATS_1 can easily mimic being a substrate for cellular LATS1, and regulate the Hippo pathway’s core components, hence restyling the defense mechanism against SARS-CoV-2 infection. Therefore, we assume that the Hippo pathway is turned off in Omicron-infected cells by diverting the cellular LATS kinase to viral MOD_LATS_1 and phosphorylating it. This step leaves the active YAP/TAZ in the cytoplasm, inhibits TBK1/IRF3, and weakens the antiviral defense mechanism. The likely involvement of MOD_LATS_1 in regulating the Hippo pathway is shown in Figure 5B. The abnormal regulation of the Hippo pathway was observed during infection with various viruses, such as HBV, HCV, MCV, ZIKV, EBV, KSHV, HPV, and MuPyV [28]. For instance, the Zika virus, a member of the *Flaviviridae* family, activates an antiviral response and the Hippo pathway, leading to the degradation of critical proviral factor YAP/TAZ [44], which could regulate the immune response and results in reduced ZIKV replication. In this context, the presence of the MOD_LATS_1 motif in the Omicron variant could potentially activate YAP/TAZ activity and subside cellular immunity. This could positively assist in Omicron replication inside cells and strengthen transmission. However, further experiments should be performed to explore the involvement of motif MOD_LATS_1 in regulating the Hippo pathway’s core components.

## 4. Materials and Methods

### 4.1. SARS-CoV-2 Protein Sequences

We obtained the protein sequence of the SARS-CoV-2 Wuhan isolate (accession number: P0DTC2) and other coronavirus spike protein sequences used in this study from the UniProt database. The mutation information of SARS-CoV-2 variants was obtained from the CDC website (https://www.cdc.gov/coronavirus/2019-ncov/variants/variant-classifications.html (accessed on 6 January 2022)). We developed an in-house Python program to generate the spike protein sequence for each variant using the mutation information from CDC.

### 4.2. Bioinformatics Analysis

SLiM analysis of the spike protein sequences was performed using the online-resource ELM prediction tool (http://elm.eu.org/ (accessed on 6 January 2022)) (Cite PubMed ID: 31680160). Multiple sequence alignment was performed using Clustal Omega (PubMed ID: 28884485), and the alignment tree was plotted using iTOL software (PubMed ID: 33885785). All data formatting, processing, and generation of the various graphs and charts were performed using in-house R scripts (https://www.r-project.org (accessed on 6 January 2022)).

## 5. Conclusions

High transmissibility and lesser infection are significant considerations regarding the SARS-CoV-2 Omicron variant. RNA viruses exhibit a high mutation rate and are equipped with various dynamic strategies that favor their rapid adaptation to the environment. Due to the significant viral evolution, SARS-CoV-2 has a high likelihood of genomic mutations that allow for these viruses to accommodate new environments, leading to never-ending long-term threats. A deep dissection of the interactions between viral and cellular proteins ultimately leads to a better understanding of the molecular bases of the regulatory networks. Considering molecular mimicry, a common strategy for coronaviruses, we identified four prominent motifs in the SARS-CoV-2 Omicron variant that could mimic and interact with human proteins. Even though this study is purely based on computational analysis, our findings contribute to SARS-CoV-2 research by providing explanations for immune hijack and evasion. Additionally, in light of emerging variants such as BA.1–BA.5, where BA.4 and BA.5 have mutations L452R and F486V in the spike protein, our study could potentially serve as the basis for more research into how these new variants can be dealt with to overcome this pandemic. Although this study puts forth a number of novel hypotheses based on sequence-level features in the spike protein, the analysis still provides an alternate angle to explaining various key elements of infectivity and pathogenicity, such as antigenic and neutralization escape. Future works must be performed to understand the precise role of these identified SLiMs, and thus obtain much deeper insight into the SARS-CoV-2 virus in the normal cell. This information could boost the development of effective therapeutic drugs to fight against SARS-CoV-2 variants.

## Figures and Tables

**Figure 1 ijms-23-08822-f001:**
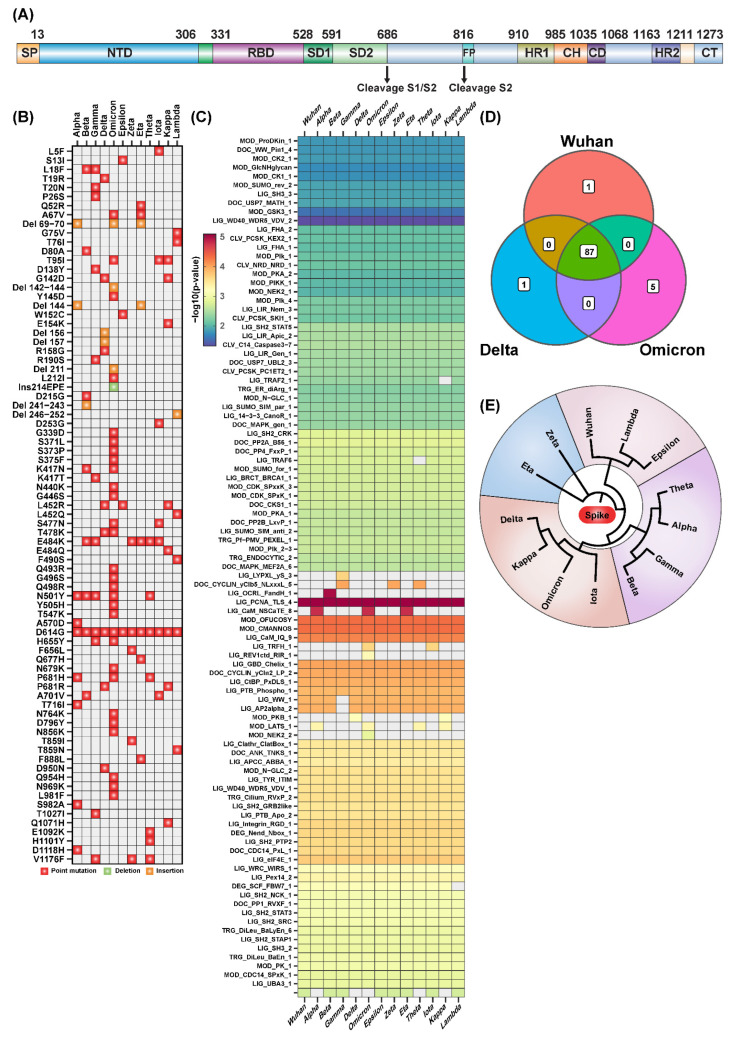
Domain organization and SLiMs in SARS-CoV-2 variants. (**A**) Protein domain organization of spike protein. The S-protein contains the N-terminal domain (NTD), receptor binding domain (RBD), subdomain 1 (SD1), subdomain 2 (SD2), fusion peptide (FP), heptad repeat 1 (HR1), central helix (CH), connector domain (CD), heptad repeat 2 (HR2), and cytoplasmic tail (CT). (**B**) Matrix of mutations on spike protein across SARS-CoV-2 variants. (**C**) Heatmap summarizing the identified SLiMs across SARS-CoV-2 variants. (**D**) Venn diagram showing the overlap of SLiMs of the spike protein from Wuhan, Delta, and Omicron variants. (**E**) A circular dendrogram showing the similarity of the spike protein from SARS-CoV-2 variants.

**Figure 2 ijms-23-08822-f002:**
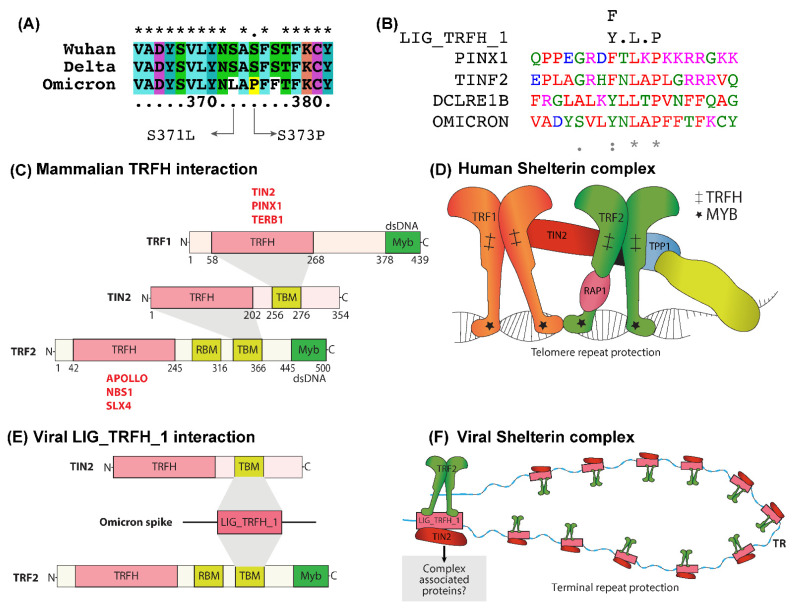
Structure and interaction networks of shelterin-complex-associated proteins. (**A**) Snapshot of the multiple sequence alignment of spike proteins from Wuhan, Delta, and Omicron, along with the specific mutations in the Omicron that led to the emergence of the novel LIG_TRFH_1 motif. (**B**) Snapshot of the multiple sequence alignment of spike protein from Omicron variant along with human proteins that contain this specific motif. (**C**) Domain organization of human TRF1, TRF2, and TIN2. (**D**) Interaction of the human-telomere-associated proteins. TIN2 bridges TRF1 and TRF2 that bind to the ds telomeric DNA. (**E**) Omicron SLiM LIG_TRFH_1 interacts with cellular TRF2 and TIN2. (**F**) LIG_TRFH_1 interaction with shelterin proteins protects the viral terminal repeats. Other proteins involved in the protective complex must be identified. TR-terminal repeats.

**Figure 3 ijms-23-08822-f003:**
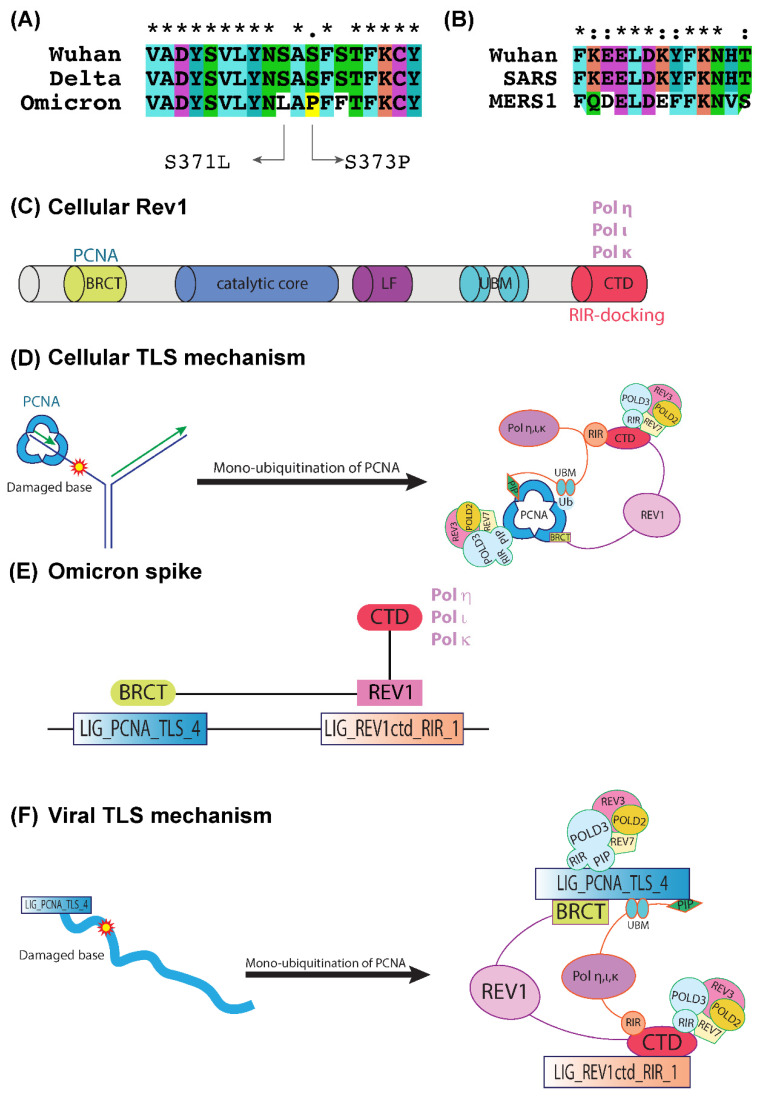
Representation of DNA damage tolerance pathway. (**A**,**B**) Snapshot of the multiple sequence alignment of spike proteins from Wuhan, Delta, and Omicron along with the specific mutations in the Omicron that lead to the emergence of SLiMs. (**C**) Schematic illustration of the domain structure of human Rev1. (**D**) Interactions of human Y-family polymerases in TLS. (**E**) Omicron SLiMs LIG_REV1ctd_RIR_1 and LIG_PCNA_TLS_4 interact with human REV1. (**F**) LIG_REV1ctd_RIR_1 and LIG_PCNA_TLS_4 motif involvement in the viral TLS.

**Figure 4 ijms-23-08822-f004:**
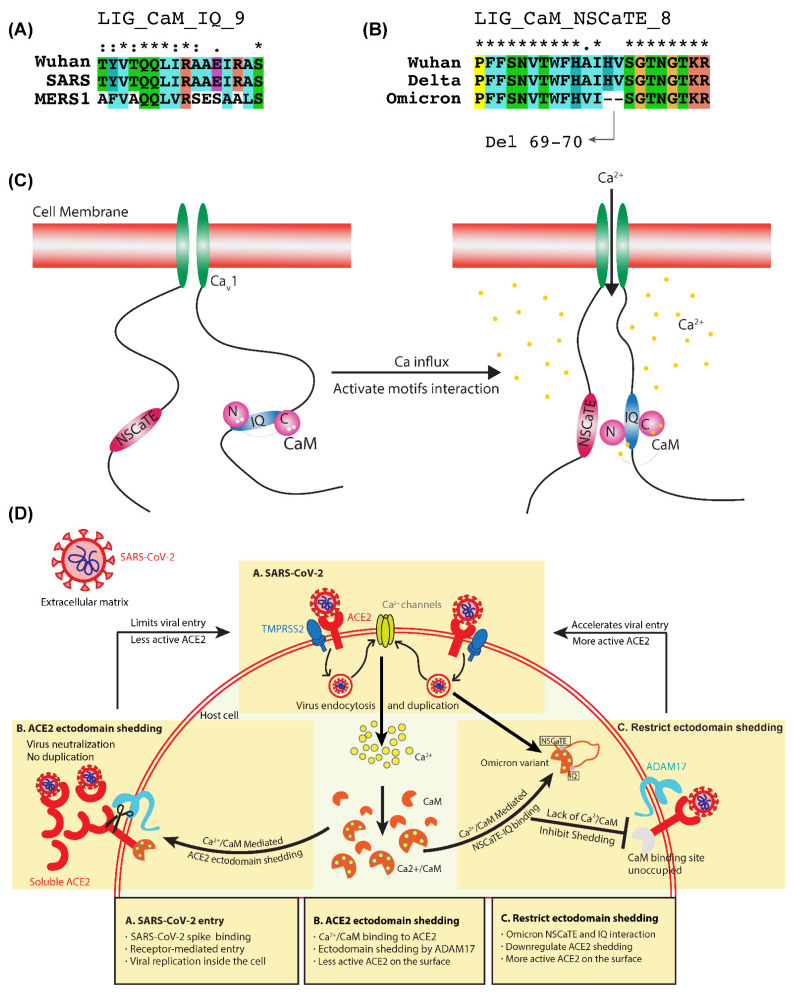
The interaction of NSCaTE and IQ motifs in a Ca^2+^/CaM-mediated manner. Snapshot of the multiple sequence alignment of spike proteins from (**A**) SARS-CoV-2 (Wuhan), SARS1, and MERS1 viruses; and (**B**) Wuhan, Delta, and Omicron along with the specific mutations in the Omicron that lead to the emergence of SLiMs. (**C**) Ca^2+^ influx facilitates motif interactions. In resting state, motifs remain unbound, NSCaTE in the N-terminus, and the IQ motif in the C-terminus. Upon membrane depolarization and Ca^2+^ influx, a Ca^2+^/CaM-complex-mediated interaction of both motifs occurs in the Cav1 channel. (**D**) A cartoon representation of the involvement of increased LIG_CaM_NSCaTE_8 SLiM-mediated transmissibility in the SARS-CoV-2 Omicron variant. The spike glycoprotein on SARS-CoV-2 interacts with ACE2 to enter the host cells. Viral entry results in an intracellular hike on the Ca^2+^ level and hence the Ca^2+^/CaM complex in the cells. Ca^2+^/CaM complex-mediated ACE2 catalytic ectodomain shedding by ADAM-17 generates the soluble form of ACE2. SARS-CoV-2 can bind to the soluble ACE2, as it contains the viral binding site, but viral neutralization occurs without an intracellular environment and cannot duplicate. When the Omicron variant enters the cells, Ca^2+^/CaM-mediated binding transpires between the unique Omicron SLiM LIG_CaM_NSCaTE_8 and the LIG_CaM_IQ_9. This process hinders ACE2 ectodomain shedding due to the lack of Ca^2+^/CaM complex availability for the CaM binding site in the ACE2 cytoplasmic receptor. As a result, more active full-length ACE2 is expressed on the surface for viral binding.

**Figure 5 ijms-23-08822-f005:**
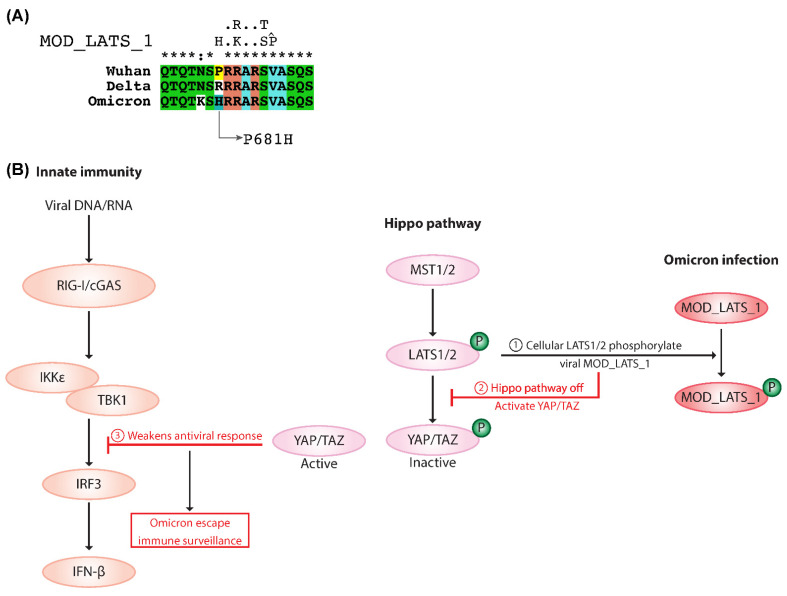
Schematic representation of the Hippo signaling pathway and Omicron MOD_LATS_1 intervention. (**A**) Snapshot of the multiple sequence alignment of spike proteins from Wuhan, Delta, and Omicron along with the specific mutations in the Omicron that lead to the emergence of the SLiMs. (**B**) Modulation in the Hippo signaling pathway during Omicron infection. When the Hippo signaling pathway is active/on, YAP/TAZ proteins become phosphorylated by LATS1/2 kinases and remain in inactive form. However, during Omicron infection, cellular LATS1/2 kinases phosphorylate viral MOD_LATS_1 leaving active cytoplasmic YAP/TAZ, which can negatively regulate immune response and facilitate Omicron survival.

**Figure 6 ijms-23-08822-f006:**
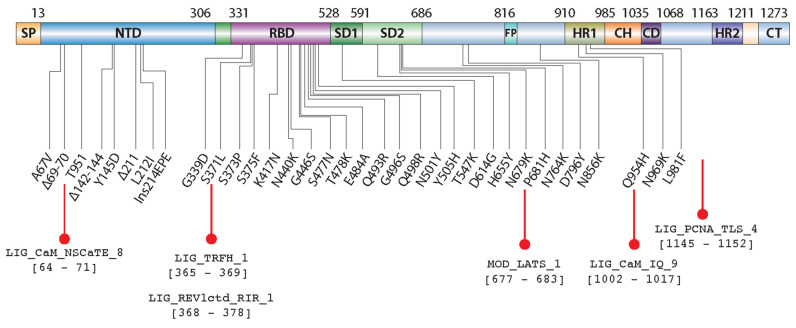
Schematic representation of Omicron spike protein organization and amino acid mutations. Omicron mutations are shown in a primary structure of SARS-CoV-2 S-protein. Amino acid mutations in SARS-CoV-2 Omicron spike proteins are A67V, Del69-70, T95I, Del142-144, Y145D, Del211, L212I, R214Insertion, G339D, S371L, S373P, S375F, K417N, N440K, G446S, S477N, T478K, E484A, Q493R, G496S, Q498R, N501Y, Y505H, T547K, D614G, H655Y, N679K, P681H, N764K, D796Y, N856K, Q954H, N969K, and L981F. Selected SLiMs introduced due to the mutation in Omicron variants are marked at the bottom of the domain map.

**Table 1 ijms-23-08822-t001:** List of unique SARS-CoV-2 Omicron SLiMs.

Viral Motifs	Accession	Instances	Regular Expression	Functional Site Class	Interaction Domain
LIG_CaM_NSCaTE_8	ELME000406	3	W[^P][^P][^P][IL][^P][AGS][AT]	Helical calmodulin binding motifs	EF-hand domain pair
LIG_TRFH_1	ELME000249	3	[FY].L.P	TRFH domain docking motifs	Telomere repeat binding factor
LIG_REV1ctd_RIR_1	ELME000450	10	..FF[^P]{0,2}[KR]{1,2}[^P]{0,4}	RIR motif	DNA repair protein REV1 C-terminal domain
MOD_LATS_1	ELME000334	23	H.[KR]..([ST])[^P]	LATS kinase phosphorylation motif	Protein kinase domain

## Data Availability

The data supporting the results can be found in Appendix A.

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
