# Peer review of "Computational Analysis of Short Linear Motifs in the Spike Protein of SARS-CoV-2 Variants Provides Possible Clues into the Immune Hijack and Evasion Mechanisms of Omicron Variant"

_ijms, 2022, doi:10.3390/ijms23158822_

Round 1

Reviewer 1 Report

The manuscript deals with the SLiMs motifs in the spike protein of SARS-CoV-2 variants. The authors identified the SLiMs motifs using bioinformatics analysis. Although the study seems interesting, the manuscript looks more like a review article than an original paper. Authors should do some experiments to prove their proof of concept or target those SLiMs using antivirals for consideration under the original paper. 

Author Response

We thank the reviewer for reading through our manuscript and providing us with their valuable feedback. We have addressed all the comments and please find below a point-by-point response to the reviewer’s comments:

Reviewer #1

The manuscript deals with the SLiMs motifs in the spike protein of SARS-CoV-2 variants. The authors identified the SLiMs motifs using bioinformatics analysis. Although the study seems interesting, the manuscript looks more like a review article than an original paper. Authors should do some experiments to prove their proof of concept or target those SLiMs using antivirals for consideration under the original paper.

We thank the reviewer for the comments. The manuscript content originates from the computational analysis of the Spike proteins followed by the potential hypotheses and discussions into the implications of the occurrence of such motifs on the infection and host response. We also agree that follow-up experiments would be a great way to address our computational results, but we will not be able to perform them due to various reasons including the lack of a BSL-3 facility as well as the required funding. We are unable to do a wet lab at this moment. Instead, we have rewritten the title and the results & discussion section, which addresses the reviewer's comments.

Reviewer 2 Report

The authors analysed the Short Linear Motifs (SLiMs) against the spike protein of SARS-CoV-2 variants. This analysis might provide a good start to further understand the characteristics of these viruses. I have some suggestions/queries to the current version of the manuscript so that if can be improved further.

- the title should be changed, since the authors did not perform experiments, my suggestion: ‘In-silico analysis of the Short Linear Motifs in the spike protein of SARS-CoV-2 variants and exploring the possible clues of immune hijack and evasion mechanisms of Omicron variant’

The organization of the manuscript was good by introducing four unique SLiMs in omicron, and then discussed them one by one.

- LIG_TRFH_1

- LIG_REV1ctd_RIR_1

- LIG_CaM_NSCaTE_8

- MOD_LATS_1

It seems that the results and discussion might be combined into one. It means that the contents of each subsection in discussions could be moved into the corresponding subsection in the methods or vice versa. The authors have to consider which way is easier to be followed by readers.

The authors have to discuss those observations should be investigated further. In addition, SLiMs could also be assessed against different omicron sublineages BA.1 to 5 in future. Finally, although all of the above were based on speculation or hypothesis stage, you can remind the readers that the analysis provides an alternate angle of explaining antigenic escape, neutralization escape, etc.

minor comments:

- there were several minor mistakes which made the manuscript not professional, the authors have to go through and polish it carefully, I listed some examples below:

(1) line 71: typo for ‘five’ SLiMs unique to SARS-CoV-2, should be ‘six’ SLiMs unique to SARS-CoV-2

(2) line 418: it is not possible to know the two human coronaviruses without defining the terms, CVHN5, CVHOC. I only realized that they referred to human coronavirus HKU1 and human coronavirus OC43 respectively after doing google search.

(3) line 84: the authors said that there were four unique SLiMs to Omicron, however, those four unique SLiMs were also shared by other variants. the term ‘unique’ was discordant to line 71. the supplementary 1A, those six SLiMs were unique to SARS-CoV-2 and did not share with other human coronaviruses.

(4) the quality of figure 1C was poor, I can only refer other figure to guess the correct names along those x and y axis

(5) Table 1 was difficult to read, try to adjust the font size, enlarge each cell, give border line

Author Response

We thank the reviewer for reading through our manuscript and providing us with their valuable feedback. We have addressed all the comments and please find below a point-by-point response to the reviewer’s comments:

Reviewer #2

The authors analysed the Short Linear Motifs (SLiMs) against the spike protein of SARS-CoV-2 variants. This analysis might provide a good start to further understand the characteristics of these viruses. I have some suggestions/queries to the current version of the manuscript so that if can be improved further.

We thank the reviewer for reading through the manuscript.

- the title should be changed, since the authors did not perform experiments, my suggestion: ‘In-silico analysis of the Short Linear Motifs in the spike protein of SARS-CoV-2 variants and exploring the possible clues of immune hijack and evasion mechanisms of Omicron variant’

We thank the reviewer for this suggestion. We have now updated the title of the manuscript.

The organization of the manuscript was good by introducing four unique SLiMs in omicron, and then discussed them one by one.

- LIG_TRFH_1

- LIG_REV1ctd_RIR_1

- LIG_CaM_NSCaTE_8

- MOD_LATS_1

It seems that the results and discussion might be combined into one. It means that the contents of each subsection in discussions could be moved into the corresponding subsection in the methods or vice versa. The authors have to consider which way is easier to be followed by readers.

We again thank the reviewer for critically reading through the manuscript and this suggestion. We have now revised the manuscript according to this suggestion.

The authors have to discuss those observations should be investigated further. In addition, SLiMs could also be assessed against different omicron sublineages BA.1 to 5 in future. Finally, although all of the above were based on speculation or hypothesis stage, you can remind the readers that the analysis provides an alternate angle of explaining antigenic escape, neutralization escape, etc.

We strongly agree and sincerely thank the reviewer for this suggestion. In the revised manuscript, the discussion section highlights that these findings should be investigated further, especially in terms of assessing against emerging strains.

minor comments:

- there were several minor mistakes which made the manuscript not professional, the authors have to go through and polish it carefully, I listed some examples below:

(1) line 71: typo for ‘five’ SLiMs unique to SARS-CoV-2, should be ‘six’ SLiMs unique to SARS-CoV-2

We thank the reviewer for pointing out these minor mistakes and we apologize for the oversight. We have corrected this specific mistake and also looked through the manuscript in detail to fix such issues in the revised version of the manuscript.

(2) line 418: it is not possible to know the two human coronaviruses without defining the terms, CVHN5, CVHOC. I only realized that they referred to human coronavirus HKU1 and human coronavirus OC43 respectively after doing google search.

We have corrected this in the revised version of the manuscript.

(3) line 84: the authors said that there were four unique SLiMs to Omicron, however, those four unique SLiMs were also shared by other variants. the term ‘unique’ was discordant to line 71. the supplementary 1A, those six SLiMs were unique to SARS-CoV-2 and did not share with other human coronaviruses.

Thank you very much for pointing this out. We have corrected this in the revised version of the manuscript.

(4) the quality of figure 1C was poor, I can only refer other figure to guess the correct names along those x and y axis

We have updated figure 1C with a high-res image in the revised version of the manuscript. We have the original illustrator file that has the vector object. In case, if the inserted image in the Manuscript is not good enough, we will work with the production team to get some help on how to add the high-res image.

(5) Table 1 was difficult to read, try to adjust the font size, enlarge each cell, give border line

We have made the necessary changes in the revised version of the manuscript.

Reviewer 3 Report

In this manuscript, Soorajkumar et al. analyze short linear motifs (SLiMs) present in spike proteins of SARS-CoV-2 variants, including Omicron, and suggest possible mechanisms for cellular interactions underlying Omicron emergence. The subject is undoubtedly of interest and importance. At the same time, it is a pity that the authors did not seek stronger arguments to support their proposed schemes.

Still, their results may serve as a starting point for more thorough research in the respective directions.

Figure 3A duplicates Figure 2A – maybe there is no need in such a duplication on two nearby pages.

The panels E and F of Figure 3, which appears in Section 2.3, are commented there very briefly, and are explicitly referred to only in the Discussion section. It might be worth saying more about that right at the time when the figure is introduced in the text.

Author Response

We thank the reviewer for reading through our manuscript and providing us with their valuable feedback. We have addressed all the comments and please find below a point-by-point response to the reviewer’s comments:

Reviewer #3

In this manuscript, Soorajkumar et al. analyze short linear motifs (SLiMs) present in spike proteins of SARS-CoV-2 variants, including Omicron, and suggest possible mechanisms for cellular interactions underlying Omicron emergence. The subject is undoubtedly of interest and importance. At the same time, it is a pity that the authors did not seek stronger arguments to support their proposed schemes.

We really appreciate the reviewer’s kind words. It is really encouraging for us! We do wish to make strong arguments as suggested by the reviewer, but we were concerned that we would be requested to provide a lot of experimental data which is currently not feasible due to the lack of a BLS-3 facility and the required funding. Again, thank you very much for the encouraging words. It means a lot for us to hear it from a reviewer.

Still, their results may serve as a starting point for more thorough research in the respective directions.

We also hope that our results and hypotheses can give new directions that can be taken up by the research community.

Figure 3A duplicates Figure 2A – maybe there is no need in such a duplication on two nearby pages.

We agree with the reviewer. However, the reason we duplicated this panel is to have all the panels corresponding to that specific result section within one figure. I hope this is OK with the reviewer. But we will be happy to remove this duplication if the reviewer insists.

The panels E and F of Figure 3, which appears in Section 2.3, are commented there very briefly, and are explicitly referred to only in the Discussion section. It might be worth saying more about that right at the time when the figure is introduced in the text.

We thank the reviewer for pointing out this issue. We have now included the explanation for figures 3 E and F in the text where the figures are introduced in the revised version of the manuscript.

Reviewer 4 Report

This is a nice manuscript displaying the potential protein-protein interactions of SARS-CoV-2 spike motifs (mainly Omicron) with the host proteins based on bioinformatical analysis. SLiMs are significant in understanding many diseases’ progression including HCC evolution in HCV and immune modulation in SARS-CoV-2.

Despite the paucity of info in this context, this type of study is of great importance for understanding the molecular bases of such a disease and possibly aiding its therapeutic antiviral development, even though further studies need to be done in order to confirm these findings.

In addition, in this manuscript the information is well structured and written, for this, I believe it’s suitable for publishing in “IJMS”.

Author Response

We thank the reviewer for reading through our manuscript and providing us with their valuable feedback. We have addressed all the comments and please find below a point-by-point response to the reviewer’s comments:

Reviewer #4

The paper brings novel and necessary info on Omicron variant and its potential SLiMs.

We sincerely thank the reviewer for reading through the manuscript and for the positive feedback.

Round 2

Reviewer 1 Report

The authors significantly improved the manuscript and it can be accepted in the present form.